# Motivations for and experiences of childbirth abroad amongst Nigerian women: A qualitative study

**Aduragbemi Banke-Thomas**[1,2]*, **Olayinka Lewis**[3], **Adeola Duduyemi**[2],
**Olakunmi Ogunyemi**[4], **Teeroumanee Nadan**[5]

1 Maternal Adolescent Reproductive and Child Health Centre, London School of Hygiene & Tropical Medicine, London, United Kingdom, 2 School of Human Sciences, University of Greenwich, London, United Kingdom, 3 Essex Law School and Human Rights Centre, University of Essex, Essex, United Kingdom, 4 University Hospitals of Derby and Burton NHS Foundation Trust, Derby, United Kingdom, 5 Association for Learning Technology, Bicester, United Kingdom

* aduragbemi.banke-thomas@lshtm.ac.uk

**Data Availability Statement:** Our article contains excerpts from the qualitative data we collected and synthesised. At the time that ethical approval was applied for, we did not specify that data would be

## Abstract

Birth tourism, the practice of a woman travelling out of her country of residence to another country to give birth, is common globally. Despite this, there is limited literature on the motivations and experiences of women who gave birth abroad. This study aims to address this gap by seeking to understand the motivations for and experiences of childbirth abroad among Nigerian women. Using purposive and snowball sampling, 27 Nigerian women who had children abroad were recruited via social media platforms. In-depth interviews were conducted remotely, audio-recorded, transcribed, and analysed thematically. Braun and Clarke's six-step thematic analysis was used, which included data familiarisation, code generation, searching for themes, reviewing themes, defining themes, and producing the report. We found that motivations for seeking childbirth abroad varied based on the mother's desires for their children, needs, and circumstances. These motivations were formed at different times before and after pregnancy and evolved over time. The experience of childbirth abroad is mostly good. However, there are also bad experiences, with some women feeling like they were treated differently because they were *"self-paying"* patients, *"black"*, or not country residents. The cost of care is deemed exorbitant, but most pay their bills. Support of loved ones around childbirth abroad was considered crucial, although not always available. Through it all, realising the expected and collateral benefits of childbirth abroad made it all worth it. <u>In conclusion,</u> motivation for childbirth abroad varies and evolves. While globalisation, broken health systems, and ongoing sustained economic challenges in Nigeria and similar settings continue to motivate women to seek childbirth abroad, their experiences of childbirth abroad suggest that though it might be greener on the other side, it is not necessarily dark green. Systems are needed to elevate their voices in the public discourse and safeguard them from bad experiences of childbirth abroad.

made available in a public repository. Therefore doing so would not be in line with the approved ethical application and the information that was provided to participants about the study and how their data would be used. Making the raw data publicly available would be a serious ethical breach in relation to the rights of the study participants. Further, given the sensitive nature of the matter discussed, it is unlikely that the participants would be appreciative of their data being made publicly available. In case there is special interest in our data, access requests can be made to the Nigeria National Health Research Ethics Committee (NHREC) at deskofficer@nhrec.net.

**Funding:** No funding was received for this study.

**Competing interests:** The authors have declared that no competing interests exist.

## Introduction

The practice of a woman travelling out of her country of residence to another country to give birth, otherwise described as *"birth tourism"*, has long existed. While there is no official agency tracking the numbers of these births across countries, an attempt to constitute an annual estimate in the United States (US) by the non-partisan research organisation known as the Center for Immigration Studies (CIS) came up with a range of 20,000 to 26,000 [1]. In the US, the practice is said to be particularly common among women from Brazil, China, Korea, Mexico, Nigeria, Taiwan, Turkey, and Russia [2]. In the United Kingdom (UK), the phenomenon is labelled by some as the *"Lagos Shuttle"*, highlighting the high number of Nigerian women said to be so-called *"birth tourists"*. At one airport in the UK, the government claimed that over a two-year period, immigration officials stopped more than 300 foreign expectant mothers at the border, with many said to have pregnancies which were too advanced, making it difficult for officials to put the women back on planes to return to their homes [3]. More recently, in Canada, a retrospective analysis of 102 women identified through a Central Triage system as women who gave birth in Calgary over 18 months and who were deemed to be *"birth tourists"* found that most were from Nigeria (24.5%) [4]. In another retrospective study of 413 birth tourists conducted in a county hospital in Chicago, Illinois, United States of America (USA), 88% of the women were of Nigerian nationality [5].

In recent years, a lot of attention has been drawn to the practice of birth tourism, which is now profoundly polarising. For example, in January 2020, the U.S. Department of State discontinued the approval of visas for birth tourism to foreigners [6]. At the time of publication of this rule, the CIS, while asking for more to be done, lauded it as minimising the number of women who could come to the US to give birth and receive taxpayer-funded public assistance to cover childbirth expenses. The CIS also described it as a means to curb access to US citizenship by *"terrorists, spies, and criminals"* [2]. In 2018, Conservative delegates in Canada voted that a child being born in the country should not guarantee citizenship [7], though this is yet to lead to any change in policy. Some obstetricians have suggested that birth tourism has only negative consequences for healthcare providers and the health system in Canada [8], with one senior obstetrician recently clamouring that care should be *"declining to accept these patients into care. . ."* [9]. In 2015, the issue also led to a political firestorm in the UK when a Nigerian woman who gave birth to a set of quintuplets in the country four years earlier was said to have been let off her £145,000 healthcare bill [10]. In 2017, another Nigerian woman who had twins in a UK hospital was said to have racked up a bill of £350,000 following a caesarean section [11]. At the other end of the discourse, women who gave birth abroad were shown to have better obstetric outcomes than resident women. However, they had challenges such as late presentation for care, limited medical records, presentation with conditions unfamiliar to care providers, and unforeseen financial obligations assumed by patients [5].

Several researchers and commentators have submitted that the appeal of foreign citizenship at birth is the main motivation for women travelling to other countries to give birth [3, 4, 12, 13]. Indeed, a recently published survey appears to support this widely held opinion, as it showed that almost 80% of respondents self-reported as "birth tourists" and stated that the major reason they came to Canada to give birth to their baby was to ensure that their child gained Canadian citizenship [4]. Beyond this peer-reviewed article, which reported Nigerian women as constituting the highest proportion of "birth tourists" (25%) [4], to our knowledge, no other study has assessed or explored motivations for seeking childbirth abroad amongst Nigerian women. Regarding experience, we are only aware of one study that engaged pregnant women who had their babies abroad to understand their experiences of care [12]. This study was conducted amongst Mainland Chinese women [12], not Nigerian women, amongst whom

the practice is common [4]. Our objective in this study is to address the gap by seeking to understand the motivations for and experiences of pregnant Nigerian women who travelled out of Nigeria to give birth in another country.

## Materials and methods

### Study design

For this research, we adopted a phenomenological qualitative study design using in-depth interviews (IDI) with Nigerian women who gave birth to at least one of their children abroad. We followed the Standards for Reporting Qualitative Research [**S1 File**] [14].

### Study setting

Nigeria, where all the women interviewed in this study originate from, is a sub-Saharan African country in West Africa. About eight million births occur annually in the country [15], yet it contributes the highest number of maternal deaths (82,000 (approximately 29%)) and the third highest number of stillbirths (171,428 (about 9%)) in the world [16, 17]. In Nigeria, a woman's chance of dying from pregnancy and childbirth is 1 in 13 –one of the highest globally [18]. Opinion of pregnant women in the country regarding the quality of maternal health services is mixed, with many women highlighting poor staff attitude, long waiting times, inadequate attention to women in labour, and substandard facilities as issues, more so in the private sector. However, many women perceive the technical capacity of specialists in the public sector as a strength of the system [19–21]. As established already, Nigeria is one of the top countries of origin for women who seek childbirth abroad [2, 4], and indeed, giving birth abroad has been described as a trend in the country in recent years [22].

### Recruitment and sampling of interviewees

We recruited Nigerian women over 18 years old at the time of the study and who had given birth to at least one of their children abroad. Those who gave birth abroad unintentionally and those who were living/resident in the country of birth at the time of their delivery outside Nigeria were excluded. Recruitment was conducted using an advert circulated via targeted social media groups on WhatsApp, Twitter, and Facebook, leveraging guidance from literature [23–25]. Through the advert accessed via a web link posted in the social media groups, prospective interviewees received a short project brief with links to the participant information sheets and informed consent forms [**S2 File**]. Those interested were requested to complete, sign, and return the consent form. To ensure women felt most free to share during the interviews, they were allowed to choose two members of the research team they preferred to conduct the interview from a list that described the background of all team members. Purposive sampling with snowballing was used in the study to reach a wide range of targeted women of interest. Efforts were made to ensure a maximum variation is represented by recruiting across various age groups, levels of education, obstetric history (singleton v. multiple gestation), mode of delivery (spontaneous vaginal delivery or caesarean section), birth country, number of births abroad (one or many), and sex of babies delivered abroad. These characteristics were selected because they have been highlighted in the birth tourism literature as relevant to motivations for or experiences of childbirth abroad [3, 4, 12, 13].

We recruited 37 women, of which 10 declined to be interviewed after initially consenting and agreeing to be interviewed. Reasons for declining the interviews included the fear of rejection for future US visas and concerns about ongoing green card applications because, despite our reassurances, some women were uncertain about the potential reach of the interview

results and the possibility of identification. In the end, 27 Nigerian pregnant women who gave birth to at least one of their children abroad were interviewed.

## Data collection

We used IDIs for data collection as they allowed for a robust exploration of each woman's experience and motivation for birth tourism abroad [26]. All interviews were conducted online via Microsoft Teams (Microsoft Corporation, Redmond, Washington, United States) or Zoom (Zoom Video Communications, Inc., San Jose, California), depending on the interviewee's preference. Irrespective of the platform used, all IDIs were conducted using an open-ended interview guide and pilot-tested with women not involved in the study. The interview guide included open questions exploring why the women chose to go abroad for their childbirth, the process of selecting the country they went to, their experience in planning the childbirth, and their experience during childbirth abroad [**S3 File**]. Efforts were made to establish rapport from the beginning of the interview to ensure the interviewees felt comfortable enough to communicate freely and behave naturally. The interviews lasted 50 to 85 minutes, depending on how many children the interviewee gave birth to abroad. Each interview was conducted by two team members, as requested by the women during recruitment.

All study team members were trained in qualitative research and ethical procedures guiding research, and they were guided in their conduct of the IDI by a standard operating protocol. All IDIs were audio-recorded, and reflective notes were taken to supplement the transcripts. The interviewers' understanding of the interviewee's comments was repeated to them to verify that what they intended to convey was correctly interpreted (confirmability) [27]. In line with good practice, the interviewer's opinions on the subject were discussed across the team and set aside before the interviewees throughout the data collection period [28]. Data was collected over four months between February and June 2023. Data collection continued until thematic saturation was achieved.

## Data analysis

All audio recordings were transcribed verbatim, and the resulting transcripts were reviewed for accuracy. For data reduction, the thematic approach, which focuses on detecting and describing both implicit and explicit ideas (themes) within the transcript, was applied [29]. Braun and Clarke's six-step approach involving data familiarisation, initial code generation, searching for themes, reviewing themes, defining and naming themes and producing the report was used in the study [29, 30]. This approach allowed us to explore contextual situations, experiences, and perceptions while highlighting important emerging themes [31]. For the data familiarisation step, all authors listened to the audio recordings and read the transcripts several times to understand the data comprehensively. For the initial code generation step, a deductive approach inspired by the interview guide was used to generate codes while allowing open coding to elicit unanticipated themes [32]. As we searched for themes, we looked for patterns and connections amongst the codes, grouping them into preliminary themes. We reviewed and refined these themes in the reviewing themes step, discussing and debating their interpretations to ensure consistency and accuracy. In the defining and naming themes step, we worked together to identify overarching themes that captured the essence of the data, defining them through detailed descriptions and examples and giving them informative names. Finally, in producing the report, the themes were organised and presented clearly and logically, with quotes and examples from the transcripts used to support each theme.

All qualitative analyses were conducted with the aid of the computer-assisted qualitative data analysis software Dedoose (University of California, Los Angeles, USA).

### Ethical considerations

Ethical approval was obtained from Nigeria's National Health Research and Ethics Committee (NHREC/01/01/2007-23/08/2022), the University of Greenwich Research and Ethics Committee (UREC/21.5.7.9) and the University of Essex Ethics Committee (ETH2122-0736) in the UK. Participation in the study was entirely voluntary and free of coercion. Each potential participant completed a written informed consent form as part of the recruitment process, guaranteeing their privacy and confidentiality. Women were free to withdraw from the study if they felt uncomfortable. The interviewers supported them if the interview became difficult by offering encouragement, giving them time to be settled enough to continue with the interview if they chose to, rescheduling the interview until the participant felt up to it, or cancelling the interview altogether. Furthermore, respondent anonymity was maintained in this study's reporting by recording their names with initials only to avoid identification. The research adhered to the highest ethical standards, in line with Beauchamp and Childress's four fundamental moral principles of autonomy, non-maleficence, beneficence, and justice [33], and ensured the safeguarding of the rights and welfare of the interviewees. Each participant was given a mobile phone top-up card valued at N5,000 (US$5) or an Amazon gift card valued at £5, US$5, or 5 CAD, depending on their current location.

### Inclusivity in global research

Information regarding the ethical, cultural, and scientific considerations specific to inclusivity in global research is included in the Supporting Information [S4 File].

## Results

Of the 27 interviewed, 23 gave birth to at least one child in the US, and four gave birth to at least one child in the UK. One woman each gave birth in Canada, Ireland, and Zambia. Thirteen of the interviewed women had all their children abroad. Twelve women had only boys abroad, including four who also had girls in Nigeria, and 11 gave birth to a mix of boys and girls abroad. All women were educated to the tertiary level. There was no clear pattern in the order of children born abroad for those who had some of their children born in Nigeria [Table 1].

Findings are presented below under five key themes:

1. Motivations for childbirth abroad vary and evolve based on their needs, desires, and circumstances.

2. Experience of childbirth abroad is mostly "good", but not in all cases.

3. The cost of childbirth abroad is exorbitant, especially for complicated cases, but most pay somehow.

4. Support from loved ones living abroad around the time of childbirth is crucial, but it is not always available.

5. The realisation of expected and collateral benefits of childbirth abroad makes it all worth it.

### Motivations for childbirth abroad vary and evolve based on their needs, desires, and circumstances

Many women stated that gaining an alternative citizenship in a country that offered birthright citizenship was the reason for giving birth abroad. This motivation was underpinned by a

**Table 1. Characteristics of interviewees.**

| Participant number | Level of education | Total number of babies born | Number of births abroad | Number of babies born abroad | Sex of babies in order of birth | Country of birth in order of birth |
|---|---|---|---|---|---|---|
| W01 | Masters | 2 | 2 | 2 | **Boy, Girl** | **US, Zambia** |
| W02 | Masters | 2 | 2 | 2 | **Girl, Boy** | **US, US** |
| W03 | Masters | 4 | 2 | 2 | Girl, **Boy, Boy**, Girl | NG, **US, US**, NG |
| W04 | Masters | 4 | 3 | 4 | **Boy, Boy, Boy & Boy** | **UK, US, UK & UK** |
| W05 | PhD | 4 | 2 | 3 | Girl, **Girl, Boy & Boy** | NG, **US, US & US** |
| W06 | BSc | 3 | 1 | 1 | Girl, Girl, **Boy** | NG, NG, **US** |
| W07 | Masters | 2 | 2 | 2 | **Girl, Boy** | **US, US** |
| W08 | Masters | 2 | 2 | 2 | **Boy, Boy** | **USA, UK** |
| W09 | Masters | 2 | 2 | 2 | **Boy, Girl** | **US, US** |
| W10 | BSc | 3 | 1 | 1 | Boy, Boy, **Girl** | NG, NG, **US** |
| W11 | Masters | 5 | 2 | 2 | **Boy**, Boy, Boy, Boy, **Girl** | **UK**, NG, NG, NG, **US** |
| W12 | BSc | 4 | 3 | 4 | **Girl, Boy, Boy & Boy** | **US, US, US & US** |
| W13 | BSc | 3 | 2 | 2 | Girl, **Girl, Boy** | NG, **US, US** |
| W14 | BSc | 3 | 1 | 1 | **Boy**, Boy, Boy | **UK**, NG, NG |
| W15 | BSc | 1 | 1 | 1 | **Boy** | **US** |
| W16 | Masters | 2 | 2 | 2 | **Boy, Boy** | **US, US** |
| W17 | BSc | 2 | 1 | 1 | Boy, **Boy** | NG, **US** |
| W18 | Masters | 2 | 2 | 2 | **Boy, Boy** | **Ireland, US** |
| W19 | Masters | 2 | 2 | 2 | **Girl, Girl** | **US, US** |
| W20 | Masters | 3 | 2 | 2 | Girl, **Boy, Boy** | NG, **US, US** |
| W21 | BSc | 3 | 1 | 1 | **Boy**, Girl, Girl | **US**, NG, NG |
| W22 | Masters | 2 | 2 | 2 | **Girl, Girl** | **US, US** |
| W23 | Masters | 3 | 1 | 1 | Boy, Boy, **Boy** | NG, NG, **US** |
| W24 | Masters | 2 | 2 | 2 | **Girl, Boy** | **Canada, Canada** |
| W25 | Masters | 3 | 1 | 1 | **Boy**, Boy*, Girl* | **UK**, UK*, UK* |
| W26 | Masters | 3 | 2 | 2 | Girl, **Boy, Girl** | NG, **US, US** |
| W27 | PhD | 3 | 1 | 1 | Girl, Boy, **Girl** | **US** |

**Footnote**: The sexes in bold and the countries in bold represent the children born abroad as birth tourists. *Children born abroad but at a time when the mother was already a legal resident of the UK.

perception that such citizenship increases their children's potential to secure future educational opportunities and a better living environment and aids easy access to future opportunities such as jobs and loans. Others also said they wanted such citizenship for their children because it helped ease travel mobility without the need for a visa or the lengthy process associated with a visa application and minimised hostilities related to landing in some foreign countries with a Nigerian passport. This motivation was reported irrespective of birth order, with some having their first child abroad if they could afford it at the time, others having their first child in Nigeria and subsequent ones in a country that offered birthright citizenship, and some like W01 had their first abroad and subsequent one in a country that did not offer birthright citizenship.

> *"The reason why I gave birth to my boy there is... You know, firstly, it is for US citizenship. Yeah, you know, everybody craves a better life for their children. Oh, and since we could afford it and it was not illegal, we went for it."* (W21, three children, firstborn in the US)

*"The US is the most powerful country in the world. You cannot mention many countries before you mention the US. And we just felt like it also gives that [citizenship], we were also trying to see how we can secure his future because when you give birth in the US, your child automatically becomes a citizen."* (W01, two children, firstborn in the US and second born in Zambia)

Women described other motivations for childbirth abroad. Some were motivated to seek childbirth abroad because they sought what was described as *"better healthcare"*. This motivation was shared by some women who had bad obstetric experiences in Nigeria, such as poor treatment by health personnel and loss of a child during previous pregnancies or those who had been advised by family, friends, colleagues at work, or other trusted persons that healthcare was better abroad. One woman (W08) who had her first child in the US and wanted to avoid giving birth to her second in Nigeria as she *"didn't trust the system"* sought *"better healthcare"* in the UK, though there was no citizenship by birth on offer. She chose the UK over the USA as she did not have a US visa, and the embassy was not yet fully open following the 2020 lockdown, but she had a UK visa.

*"When I was pregnant with my first child, the one I lost, my brother-in-law was in America as at then, and he was saying come over, come, and have this baby here [US] and all that. I wasn't in for it. I was like, no, I'm not interested in stuff like that. And then eventually, we lost the baby after three weeks of birth due to our Nigerian factor [Issues in the Nigerian health system], you know. That made us reconsider going abroad to have the other ones so that we can get the best medical attention that we needed because it was due to laxity and the mistake on their part that we lost that baby."*–(W03, four children, first born in Nigeria as stillborn, second and third born in the US)

In addition, some women said they were motivated to seek childbirth abroad because that is where they had family who could support them during the *"difficult period"* of childbirth and the initial few weeks after their baby is born. One woman (W14) had her first child in the UK because her spouse was there then, and she wanted to have him by her side. For some women, the motivation was driven by a combination of the abovementioned factors. For older women over 35 years at the time of the pregnancy, better healthcare seemed to be their primary motivation as they did not want to take chances with their health or their lives.

*"I was fine with my first two pregnancies in Nigeria. Honestly, I was really fine. The third was a case in which nothing must go wrong with the baby or me. It was not a planned pregnancy, and now I was older, and I really needed to get away because it was strange being pregnant at my age. I remember there was a colleague at work who had lost his wife, who was pregnant at an older age around that time, and that worried me. . . So, I had to go to where I had family and where I could be comfortable and get the support that I needed. . .".*—(W27, three children, third born in the US)

*"We decided we were going to have the first in the UK. The primary reason was better health care, and then the cherry on top was the fact that you know my mother was in the UK. . . The reason for the US for my second son was a couple of things. So, number one, my mum [who was living in the UK] had sadly passed away at the time. . . So, we had decided we would go to the US, as my sister was there, so there was another support in the US."*–(W04, four children, first, third and fourth born in the UK, second born in the US)

In terms of how motivation was formed, it appears to be established for some women way before pregnancy, while for others, it was during pregnancy, as an emergency. There was no clear pattern regarding which women established their motivation before or during pregnancy. However, some women reported that motivation sometimes evolved over time based on their perception of risk. The decision to seek childbirth abroad was formed by women, usually in agreement with their partners or encouraged by others.

*"So, having a second citizenship was something we had discussed even before we married. We had planned towards after we got married or, let me say when I became pregnant."*–(W09, two children, first and second born in the US)

*"I think it's funny because when I was pregnant, my boss just told me, why don't you go have your children abroad? There are more opportunities for them [abroad], and that's why [we did it]. I got home, discussed it with my husband and boom, we decided to do it. . . It wasn't a well-thought-out plan."*—(W22, two children, both born in the US)

## Experience of childbirth abroad is mostly "good", but not in all cases

Generally, women were satisfied with their birth experiences abroad. However, a few women felt the care was the same as they would get in Nigeria, while others felt that the care they received abroad was not as good as expected. Most women reported receiving *"good care"* from hospitals abroad, which was described as care that involved painless vaginal delivery with epidural anaesthesia, even when this came at an additional cost, comfort throughout the labour, and the availability of necessary equipment for childbirth with good preparedness demonstrated by skilled health personnel to manage any complications. A handful of women who had given birth in the US and the UK compared their experiences in both countries, noting that in the US, they felt they had access to different cadres of professionals during childbirth, and it was a more pleasant experience than in the UK. However, the UK has a post-childbirth visitation service, which was not available to them in the US. Experience also appears to vary even for the same woman across different births, with one woman who had her two babies in the US (W02) saying, *"I have had multiple experiences in the US, the good, the bad, right?"*. It appeared that the good experience was service-specific and not specifically related to the country of birth, with a woman (W01) who had one of her children in Zambia, an African country like Nigeria, reporting that she received "good" care there.

*"In the US, I had two or three people [health personnel] flanking me. The doctor was there. I was on the bed. You could see the technological advancement from where I came from [Nigeria]. Everything was monitored almost every minute. . . and they kept explaining the stage I was in, kept asking me if I was ok."*—(W20, three children, second and third born in the US)

*"In Zambia, it was the same [with the US], the doctor was present. . . I was also quite impressed they still insisted on showing me the bathing process again, like how to bathe the baby, but they did not give as much information as I got in the US, but I think it was because it was not my first experience [in the US]. But when it comes to their service, the room we also stayed in, in fact, that was one thing that gave me a concern [for Nigeria]. If an African country would also have such, what are we in Nigeria doing? The workers were very polite and friendly and, like I said, provided quite similar to the service we got in the US. One thing about the room is that when you look around, you can see this ready-to-go equipment. So that if anything goes south, they are there. I was pretty impressed."*—(W01, two children, firstborn in the US, second born in Zambia)

 

*"I used [a hospital in Texas], and the facility was beautiful and very organised. They start from the moment you step in till you leave. They are very caring, and the security was top-notch, too. They had everything. There was a chapel and a cafeteria, and you know, they took everything into consideration. My experience of having my child in the hospital in the US was a fantastic, amazing one, but there's no aftercare, so once you have the baby, you're literally by yourself. Now, the experience of having my child in the UK is the opposite of that for me; having my child in the hospital, the experience was horrible. But then the UK has aftercare. So, somebody is coming to say, hey, how is [baby] doing? Do you have any problems and things like that?"*—(W07, two children, both born in the US)

Some women who had given birth both in Nigeria and abroad compared their experiences, noting what they described as a significant disparity in the care they received, with most being more satisfied with their experience of care abroad compared to that obtained in Nigeria. In addition, women mentioned that the healthcare personnel abroad demonstrated a higher level of professionalism, gave better bedside care, and appeared more empathetic than those in Nigeria. However, a few women noted that doctors and nurses who had supported them during childbirth in Nigeria were equally technically sound and that there was not much difference in how they carried out birthing procedures, though they were said to be limited by their work environment and equipment availability.

*"We have good nurses and doctors back home [in Nigeria] who are very careful. It's just that they [health personnel in the US] go the extra mile here, and they show you that respect. They don't see it as if they're doing you a favour. In the US, they take it, as you know, it's their job, and you see empathy in them, which is not as common as what we have back home."*—(W06, three children, last one born in the US)

*"The swiftness of the medical personnel, I don't think I can ever forget that in my life. It was different from what I'm used to in Nigeria, though, so I can't say and can't compare it to other hospitals in the US or what I'm supposed to expect, but the care was great, and you know I had complications, so maybe because of the complications, the care was more prompt."*—(W22, two children, both born in the US)

*"[Health personnel in Zambia] had this constant monitor, and I think some hospitals in Nigeria are beginning to have it also. So, they could tell if the child's heart rate was dropping or that of the mother was being elevated."*–(W01, two children, firstborn in the US, second born in Zambia)

Women had very high expectations of the quality of services in the countries where they sought childbirth abroad. As such, when the service quality did not meet these expectations, women were disappointed and felt they did not get a good return on their investment. Women also reported issues related to feeling treated differently from others because they were *"self-paying"* patients, *"black"*, or just not seen as native to the country. Despite the bad experiences that some of the women had, they chose not to report, query, or legally challenge the hospital because they did not want issues in a foreign land or anything that could affect their subsequent plans to have their other children abroad.

*"She [My sister-in-law] was fighting them for me because it was infuriating. Somebody was in pain, and the nurses were just going up and down. Nobody even checked to ask how I was feeling. It was only after I had spent four hours in the emergency room that I got the epidural. Then they now did the whole C section. It was not a good!"*–(W16, two children, both born in the US)

"...There were some things I did not appreciate. For example, sometimes, I didn't have good interaction with the doctor because he knew I was a self-paying patient. I wasn't an insurance patient who he needed to treat nicely. So, there were times when I also learned that I needed to bargain. I didn't just come with oh, you just tell me oh this is the price, and I will just keep quiet"–(W05, four children, last three born in the US)

"In the UK, I remember I pushed my baby out; I heard oh, you need to take a bath quickly, you need to go back to the ward because somebody else needs to use this room and I remember thinking, oh, this is the worst... and you know. I don't think a doctor ever came. It was only the midwives... I don't think my midwives in the UK were friendly in my birthing process. I remember somebody snapping at me... I feel like if she had been black, maybe they would have been nicer."–(W08, two children, firstborn in the US, second born in the UK)

"Whites mostly dominate Michigan, and so first, you know, I didn't feel very comfortable with the fact that I absolutely didn't see any black nurse."—(W10, three children, last born in the US, tried to give birth to older children in the US too, but it was not possible)

### The cost of childbirth abroad is exorbitant, especially for complicated cases, but most pay

Many women talked about the exorbitant cost of care that they had to pay, and this was a major consideration, especially when this had to be converted from Naira to foreign currency. However, almost all women felt the exorbitant cost was worth it for the expected benefits. Some chose to go to a country where it was less expensive to give birth to their babies. For example, a woman talked about choosing Canada over the US for this reason. The funds came from savings or loans they took in Nigeria or their relatives and friends in the birthing country. Some women also talked about raising funds through cooperative thrift donations, which also meant a lot of self-denial about what they could have otherwise spent their money on. For women who gave birth in the US, a few reported that they incurred unexpected/unexplained expenses during their admission in hospitals abroad, which they then had to either negotiate with the facility or agree to pay in instalments so they could obtain a zero-balance receipt to avoid subsequent denial of visas.

"...the third and the fourth were twins. That was really stressful because the costs just sky-rocketed at that time. Financially, we weren't ready. It was really a lot."—(W05, four children, last three born in the US)

"I would say it was relatively very significant when we converted to Nigerian currency... So, the first one took maybe about $6,000 or $7,000. I think the second was about $10,000 in 2012. So, I came from the house and I came to clear my bills. I went to Marketing, and I asked if there was any discount. And he said, oh, yeah, sure! Are you paying once, or do you need financial aid? I said I don't need financial aid. I'm capable of paying. I planned for this pregnancy, blah, blah, blah... And there and then I was given I think, a 30% discount... I paid cash. But I know a friend who had twins. Her baby was about $30,000. She was given a 40 or 50% discount, and she was also going to pay cash. This was in 2016."—(W20, three children, second and third born in the US)

"It came with a lot of self-denials [of things you want]... all my àjɔ̀ [thrift donation]. My husband was doing his own, too. I wasn't buying clothes like throughout those years of childbearing. It was as if we didn't have any other thing to do, we just put our finances into it."—(W12, four children, all born in the US)

*"We had access to the US but comparing the cost of childbirth there with Canada, the Canadian one was fairer, and we were happy with it. So, we went for it."*—(W24, two children, both born in Canada)

## Support from loved ones living abroad around the time of childbirth is crucial, but it is not always available

The women interviewed had varying experiences of support from loved ones during their time abroad. Most women stated that they travelled to give birth abroad alone; some went with their young children, who they could not leave behind in Nigeria as there was no one to cater for them and their partner, or in a few cases, with their mother or mother-in-law, and this determined how much support they received while abroad. At the same time, some had their spouses join them just before or after childbirth, and they were glad that they had this support. In one instance, a woman (W02) recounted how efforts had to be made to seek out distant relatives who could support her when she developed a complication after childbirth when she had not travelled abroad with any support from Nigeria.

*"I'm glad I had my husband and my cousin with me. Even when my cousin had to go to work, my husband was there, you know, because I had my baby through caesarean section. My husband had to do the bathing of the baby and, at times, food shopping and all that stuff for those few weeks that I was not myself. . . and besides that, the emotional support, the psychological support, you know, you need that, being alone in a foreign land is not always easy and considering the kind of society or culture we have back home where we have, you know when you put to bed, you've got people around you, you hardly, you know, lift your finger."*—(W06, three children, last one born in the US)

*"For my daughter, right? Six days post-childbirth, I had postpartum haemorrhage. It was the scariest thing I've ever seen. . . And so, during those periods, I was in the hospital alone. My husband was supposed to come, but at the same time, there was something they were doing in his family. My mother couldn't get her international passport ready, so I was just alone in the US. It was then that my husband started making calls, and we realised that one of his distant cousins was in another city. So, we were both in Texas, [but in different parts of the state]. So, she had to travel almost 45 minutes away to come and stay with me that period, you know."*–(W02, two children, both born in the US)

Accommodation and transportation were key support needs for Nigerian women who had babies abroad. A number of women specifically highlighted that a place to stay and someone who could help them commute back and forth the hospital informed their decision on the specific place they stayed when they travelled to give birth to their babies abroad.

*"Well, for me, I had to look for where I had family. You know, going to the hospital by yourself is not good. Anything can happen. You need someone who is family to be with you in the hospital. So, for me, I had to just go to a place where I know I have a family."*–(W06, three children, last born in the US)

*"So, I had to stay with my cousin. She has her family there. She had a home. . . It was easier for her to help me with transportation to and through the hospital. Being a mother herself, she was able to take care of me, and accommodation was sorted out for me."*–(W22, two children, both born in the US)

Some women who stayed with relatives or friends reported feeling supported by them, even when their spouses were not abroad with them. Others noted that the people they stayed with were either unavailable to support them due to the nature of their work or left them alone to sort themselves out because of the notion that *"everyone did things on their own abroad"*. Almost all women emphasised the importance of having a sound support system abroad to reduce the stress of having a baby on the mother, to reduce the incidence of depression, and to avoid rushing back after childbirth. Some of the women who did not have support mentioned that they had to deal with stress, anxiety, and shortage of funds, all of which added to the stress of being a new mother. An experience that some said was unlikely to have happened if they had given birth to their babies in Nigeria, where they had their usual social support network.

*"In 2014, I stayed with my family friend. They were great! My husband's mom was also with me, so we were together. She was the one that, you know, was with me pre and post the birth, and so it really was a great help for me."*—(W07, two children, both born in the US)

*"Well, for me, looking back at when I gave birth to my baby abroad, you know I shouldn't have stayed with anybody. . . I didn't enjoy my time with the family I stayed with. . . I should have stayed in an Airbnb or somewhere."*—(W18, two children, first in Ireland, second in the US)

*"So, my husband wasn't with me [when I came to give birth]. We didn't plan for him to come because it was a last-minute decision [to give birth in the US]. Someone just advised me that with my complicated pregnancy history, I should travel abroad, and within a week, I came. And yeah, the difficulties were more about where to live and day-to-day transport to the hospital because I was spending a lot on Uber. When the person I was staying with was to carry me, there were always a lot of issues. So, I had a nasty experience in terms of living conditions [while I was abroad]."*- (W05, four children, last three born in the US)

*"Then this other person [who I stayed with in the US], they had three children, and I had to carry my baby for three months. Nobody helped me. . . They will tell you this is America; everybody is busy, and you have to do things yourself."*–(W11, five children, firstborn in the UK, last born in the US)

One woman who had two children, one in the US and the other in Zambia, raised concern relating to postpartum depression (W01), saying, *"Postpartum depression is real because, a couple of people that have travelled abroad, they suffered with that a little bit bearing in mind that when they travelled abroad, have their child, it's more like, especially for new mothers, they are stuck with this baby crying morning, afternoon, and night"*. However, having support and someone to call to share the burden certainly helped, with another woman (W03) who had two of her children in the US saying, *"So, if I'm in that place [of stress], I will call him. . . And then, at a point, his cousin had to call me to tell me that I shouldn't be calling him because he's not here with me, and he would feel that it was really bad, worse than it really was. . . I didn't fall into depression because if I needed to cry, I cried; if I needed to yell, I yelled"*.

## The realisation of expected and collateral benefits of childbirth abroad makes it all worth it

Many women said they had already realised the benefits of giving birth to their children abroad and expect to continue to harness more benefits as their children grow older. They can use the opportunities available due to their status as dual citizens. For some, the benefit was experienced immediately after childbirth abroad, with a few women saying that they would

have died from pregnancy and childbirth complications without the care they received abroad. Others said they had experienced the benefits when travelling with their children, as they did not have to experience long delays in visa applications or at the point of entry when they arrived in certain countries. These women talked about collateral benefits for themselves since they also did not need to wait for so long because they had children with citizenship from elsewhere. In addition, some women who had older children who they had abroad could speak about the realisation of those better opportunities, exposure, and almost free tuition that their wards have enjoyed for higher education.

> *"When I came out of the theatre and opened my eyes, I noticed that the doctor was just beside me. And then he said, oh, you were crying all through your surgery, right? And then said was whatever you paid to come here was worth it because you lost a lot of blood. . ."*–(W02, two children, both born in the US)

> *"I can say we are harvesting those benefits now. The first one graduated not from Ireland but from Canada. But the second one, of course, we had to let him go to the US so that he could benefit from the reason we made that decision in the first place. For the one in the US, we're practically paying nothing. . . Even whenever we have to travel with their passport, we have not needed to go and queue for visas. So, we have had all those good benefits, because everywhere we went as far as you show the blue passport, the door Is like open to us so to say"*– (W18, two children, first in Ireland, second in the US)

Of those who had children in Nigeria and abroad, a few women believed that their children born abroad were doing better or better placed than those born in Nigeria, even though they also had a good education in Nigeria. Some women also flagged that though at the time they had their children abroad, they had no plans to relocate themselves, the more recent economic situation in Nigeria prompted them to reconsider their position and, in such instances, their children who were born as citizens in the country they emigrated to settled much quickly and were able to benefit more than those born in Nigeria. For others, the benefit was that they now had an alternative country to seek refuge considering the state of the world, though this was not the case when they sought childbirth abroad.

> *"It [childbirth abroad] was worth every dollar, every Naira, every stress, and everything. It was worth it because I've relocated and live in the US now. And the fact that I had them here, I see the huge difference between the quality of life and the resources available to all four. But I see the difference in what is available to the three born here compared to my daughter, who is not an American citizen yet."*–(W05, four children, last three born in the US)

> *"The kids are not grown yet, but I consider it a good return on investment, to be honest. . . You know the way that everything is right now in Nigeria and many parts of the world, and I will certainly do it again."*–(W08, two children, firstborn in the US, second born in the UK)

Some who talked about future opportunities suggested they were happy to delay the benefits as they would prefer to raise their children in Nigeria until they reach the age of college or secondary school. In their opinion, this will ensure that their moral, religious, and cultural values are well-formed before the children travel abroad for opportunities such as university education where they believe *"quality is guaranteed"* and *"the government highly subsidises school fees"*.

> *"Ok, so for me, it's opportunities in the future, maybe by the time they get to 18 like I don't want them to school in the US till maybe they want to go for their masters or BSc. I want them*

*to have their primary and secondary school education here in Nigeria, build that solid foundation and discipline, and have the ability to make decisions and think right. Then after that, if they want to do their BSc, master's, or PhD in the US, that is fine.*"–(*W02*, two children, both born in the US)

## Discussion

### Summary of key findings

We set out to understand the motivations and experiences of pregnant Nigerian women who sought childbirth abroad. We found that motivations for seeking childbirth abroad varied based on their desires for their children, needs, and circumstances. These motivations tend to be formed at different points in time before pregnancy and evolved over time. The experience of childbirth abroad is mostly good. However, there are cases described as bad with women feeling like they were treated differently because they were *"self-paying"* patients, *"black"*, or not native to the country. The cost of care is deemed exorbitant, but many pay their bills by some means. Support of loved ones around the time of childbirth abroad was considered crucial, but for different reasons, it was not always available. However, through it all, women said realising the expected and collateral benefits of childbirth abroad makes it all worth it.

### Interpretation of findings

Our findings show that there are a number of motivations for seeking childbirth abroad beyond the widely reported desire to gain foreign citizenship [4]. In our study, it is clear that this desire was underpinned by both perceived and experienced gains of this foreign citizenship, including the perceived increased potential of their children to secure future quality educational opportunities, better living environment, and easier access to jobs and loans. Another motivation cited by women in our study was to benefit from *"better healthcare"*, especially for those who have either had bad experiences during previous births in Nigeria or are concerned that they are carrying a particularly high-risk pregnancy. Ghanaian and Chinese women have stated similar reasons relating to birthright citizenship leading to a secured future and better healthcare as their motivation for seeking childbirth abroad [12, 13, 34]. About 60% of urban dwellers in Ghana have a favourable disposition to childbirth abroad. In contrast, Ghanaian women who had babies abroad have a perception that a US citizenship is *"more beneficial because of the assurances of certain rights and benefits that Ghanaian citizenship does not guarantee"* [35]. Some in the country refer to foreign citizenship as an *"advancement strategy"* [13]. This is similar in some respects to what we found in our study with Nigerian women. This perception is also established within the Nigerian community and appears to be perpetuated and reinforced by the testimony of others, traditional and social media, and experience [36]. One perception that we could not fully establish from our study was if the desire for foreign citizenship was a status thing, such that it was simply sufficient to be able to say one has a child born abroad, as women disagreed with such supposition even when it was put to them. It seems they were more motivated by the expected benefits of such citizenship for their children. However, it also appears that for some Nigerian women, the desired value for seeking childbirth abroad is in dual citizenship and not necessarily foreign citizenship alone, as they believed the Nigerian citizenship could help their children inculcate *"morals"* while gaining the *"opportunities"* that foreign citizenship offers abroad. With local commentators now expanding awareness that foreign citizenship could also be a disservice to children born abroad because of the potential double taxation they face as dual citizens [37], a more nuanced view of the value of

foreign citizenship by birth might be emerging. In a 2020 article, Olorunnisola concluded that childbirth abroad for Nigerians is not an *"investment"* but simply a provision of an *"option/ opportunity of living in a supposed 'better environment'"* for the child [22]. The other commonly reported motivation for better healthcare was either standalone, as there were women in our study who gave birth abroad in countries that did not offer citizenship by birth at the time of delivery or combined with the desire for their children to have foreign citizenship at birth. Women justified the motivation for better healthcare as being because they were carrying a high-risk pregnancy, had/been told about poor previous experience, perceived poor capacity of Nigerian hospitals, and had knowledge of poor pregnancy outcomes in Nigeria. The country contributes the highest number of maternal deaths, one of the highest number of stillbirths, and one of the worst experiences of maternity care [16, 17, 38].

One other motivation highlighted by many of the women in our study and which we have not found reported in the literature is that they were also motivated to seek childbirth abroad because this is where they had loved ones to support them through the precarious period of pregnancy, childbirth, and postpartum. Indeed, the number of Nigerians living in the US has increased over time, and as per recent estimates, more than one in 10 African immigrants in the US are Nigerians [39]. In a 2019 paper, Oyebamiji and Adekoye described this increase as *"not a new phenomenon nor one born out of desperation"* [40]. However, it might be challenging to make the same assertion with the current pace of emigration out of the country to other countries which do not even offer citizenship by birth but a pathway to leave the country in the first instance [36]. As of 2018, 45% of surveyed Nigerians said they planned to move to another country within the next five years–the highest proportion of the three African countries surveyed [41]. Clearly, motivations for childbirth abroad vary for women in our study, and others have been published in the literature. For example, in addition to similar reasons such as the ones highlighted by Nigerian women in our research, Chinese women reported that they wanted to give birth to their children abroad as a way to circumvent China's one-child policy [12]. Also, in an article published in The Guardian, about 66,000 Americans had their babies abroad in 2019 to avoid the extremely high cost of giving birth and starting a family in the US. However, it is not clear why the article described this practice as *"reverse birth tourism"*[42] since *"birth tourism"* alone captures what the American women did–go abroad to have their babies [42]. Some Canadian women living on the US-Canada border have also been motivated to give birth to their babies in the US because the hospital in the US was closer to them than the one in Canada (co-called *"border babies"*), thereby gaining dual citizenship by birth [43]. There is always a reason for seeking birth abroad (the 'why?'), and the choice rests with women and their partners on where they choose to give birth to their babies, provided it is done within existing legal remits.

In terms of the experience of childbirth abroad, although most of the countries considered by the women in our study generally had good reputations for their health systems, women reported varied experiences, mostly good but some bad, and some even had different experiences in different countries, hospitals, and on different birth tourism trips. The good experiences appear to cover the spectrum of expected quality of care, including the technical provision of care and the experience of care itself [44]. Some of these care features that constitute a good experience of care were also described by Ghanaian women who gave birth abroad [13]. In our study, women who had given birth in different countries were able to provide some comparative assessment, which appears to align more with how childbirth services are delivered in the countries. However, there was a range of experiences from extremely satisfied to dissatisfied/disappointed, as has been similarly reported by Mainland Chinese women who sought childbirth in Hong Kong [12]. Most of the dissatisfaction expressed by Nigerian women who sought childbirth abroad stemmed from a feeling of being treated differently

because they were *"black"*, *"self-paying"* patients, or not native to the country. It is important to highlight that discrimination while using maternity services has been reported by black women native to the most common foreign countries of birthing observed in our study–UK and US–and in the US, when pregnant black women do not have health insurance [45, 46]. As such, it is more likely that these experiences are a reflection of broader issues in the foreign health systems and not targeted intentional maltreatment of Nigerian women. Comparatively to their experience of childbirth in Nigeria, women in our study who had given birth in Nigeria and abroad highlighted that there were some positives in their Nigerian experience but adjudged their foreign childbirth experience as better. Indeed, within Nigeria, childbirth experiences vary, with positive points being the technical depth in many tertiary public hospitals and more respect for women reported in some private hospitals. However, generally, quality of care remains a concern [19, 20, 47]. In a four-country facility-based cross-sectional survey conducted in Ghana, Guinea, Myanmar, and Nigeria, which involved continuous observations of labour and childbirth done from admission up to two hours postpartum, Nigeria had some of the highest rates of mistreatment of women during childbirth with almost 70% experiencing some form of physical abuse, verbal abuse, stigma or discrimination [38].

Another key theme emerging from our study relates to the cost of care. As in the Ghana study [13], many women in our study said they saved up or loaned money to be able to pay for childbirth abroad. Even though care cost was a significant concern for most women, especially as childbirth cost was significantly less in Nigeria compared to what is obtainable in countries like the UK, US, and Canada [48–50], the vast majority were able to pay off once, in instalments, or negotiated rebates with/received discounts from hospitals abroad. However, some women incurred unexpected huge costs due to complications that resulted in caesarean sections or other surgical procedures, subsequently increasing their bills, which they struggled to pay back. Such unforeseen bills have previously been reported amongst birth tourists in a US hospital [5]. Our findings on many women paying align with those from a recent study of birth tourists in a Canadian hospital, which showed that the majority (96 of 125 (76%)) had no outstanding bill after discharge. However, the remaining 29 women had a total of $290,000.00 of outstanding fees after discharge (average $9,704; range $948-$72,445) [4]. Medically uninsured residents also have outstanding bills to pay, so this is not unique to those who have given birth to their babies abroad [51]. However, it is understandable why such debts will be a matter of concern for health personnel/service providers in the birthing countries [8].

Another important finding from our study was the criticality of support from loved ones, including family and friends, while abroad to give birth to their babies. The ideal support appears to involve supportive loved ones from Nigeria who travel with the woman to the birthing country and others who reside in the birthing country already. However, this is not always the case, as partners and relatives living in Nigeria have to manage other commitments or are hindered from travelling for different reasons. We are not aware of similar reports in the literature. For the support available in the birthing country, it is not entirely surprising that many Nigerians have people they can connect with to support them while abroad. Like the qualitative component of the mixed methods study of Ghanaian parents who gave birth to their children abroad, all in our study had some tertiary education (from post-graduate diplomas to doctorates [35, 52]. To have this level of education, it can be argued that the women are middle to high-income and would most likely be connected to a broad network, have family or friends who have settled status in foreign countries, and have access to funds to support them while having their babies abroad. However, though such support may be available without much hassle for many, others have to search hard to find such support in the birthing countries. In-country support appears to be more critical, especially when support from relatives living in Nigeria is not available. When such dependable in-country support is not available, the

experience of childbirth abroad can be challenging, with potentially negative implications on the health of the mother. As in our study, some Ghanaian women reported that they have been poorly treated by friends and family who have hosted them abroad, and, in some cases, relationships with in-country support have broken down [13].

## Strengths and limitations

One strength of this work is that, for the first time, Nigerian women have been able to narrate their motivations and experiences of foreign childbirth. Further, this study included women who gave birth in a number of countries abroad, including the UK, US, Canada, Ireland, and Zambia. In addition, by recruiting women with varying demographic and obstetric characteristics and backgrounds, we were able to maximise the heterogeneity of the sample. However, despite our best efforts, most women in this study gave birth in the US, which could be a limitation. However, it is unlikely that the high proportion of births in the US seen in our study is different from what occurs in reality since the US is seen as the leading country for childbirth abroad, which still offers citizenship by birth [22]. In addition, we cannot entirely rule out that there are no other differing or unique experiences that we have not captured. However, we continued data collection until thematic saturation was achieved in line with good qualitative research practice [29, 30].

## Implications for policy, practice, and research

First, efforts should be made to strengthen the capacity of skilled health personnel in Nigeria in terms of both the technical aspects and experience of care within the public and private sectors. Promotion of good practice in media should be encouraged, as the negative birth stories are drowning the positive ones in Nigeria. If women choose to have their babies abroad, skilled health personnel in Nigeria should discuss intentions to seek childbirth abroad with women and plans for support as much as possible. Established linkages with providers abroad might help facilitate better experiences for women when they arrive abroad.

Revisiting the motivations for childbirth abroad that women said in our study, our findings did not suggest that this was any different whether the parents were expecting a boy or a girl. We also could not say if the level of education influenced motivation as all the women in our sample were educated to at least an undergraduate level. However, obstetric history and perception of the risk of the current pregnancy appear to play an important role in motivating the choice to seek childbirth abroad. It was certainly not in our place to judge the veracity of their motivations; in any case, that was not our objective in this paper. However, it is fair to say that they all came across as germane. Even in countries in which commentators have pushed back against birth tourism, women from those countries also give birth in foreign countries. It is not about the motivation of the women, as women have a choice to give birth to their babies, provided this is done within their legal remit.

On the side of the receiving countries, as long as birth tourism remains legal in the country, then the focus needs to be on regulating it and elevating the voices of women who give birth to their children abroad rather than denying them care [9]. Women who have legally gained entry into the country should expect *"good"* care and should be able to report bad experiences and hold health personnel involved in bad practices accountable. As the topic remains a political mine bomb worsened by the rhetoric around birth tourism, especially in countries that offer birthright citizenship and free healthcare at the point of use, [2, 8, 10] there is a need to reconsider the tag *"birth tourist"*, which trivialises the situation that some pregnant women who give birth abroad find themselves. Many cannot be referred to as *"tourists"*, which Cambridge Dictionary are *"person who travels and visits places for pleasure and interest"* [53]. Such

a tag stigmatises pregnant women who seek childbirth abroad and negatively influences the perception of members of society, including skilled health personnel, which might affect their care provision. As Freckelton expertly argues in their article titled 'Could Canada Abolish Birthright Citizenship?', there is no legal basis to describe so-called *"birth tourism"* as *"an abusive and exploitative practice"*, *"loophole"* or *"fundamentally debasing the value of Canadian citizenship"* [54]. We would argue that rather than referring to them as *"birth tourists"*, the term *'cross-border birth seekers'* or *'cross-border maternity seekers'* should be used as it better characterises all pregnant women who give birth abroad. Finally, existing laws, such as the Emergency Medical Treatment and Labour Act (EMTALA), which mandate hospitals participating in the Medicare programme to provide appropriate screening to any patient seeking emergency services, and if an emergency condition exists, the hospital must either stabilise or transfer if stabilisation is not possible [55], provide an opportunity for policy action. For a pregnant woman, stabilisation means delivering the foetus and the placenta [56]. EMTALA can be better refined to establish not only the provision of care but also the experience of care. Legitimate concerns such as recouping funds from cross-border birth seekers who do not pay their bills can be addressed through other means [54].

Regarding future research, there is a need to explore the travel experiences of women who sought childbirth abroad in detail and conduct surveys to allow estimation of the magnitude of some of the issues identified in this qualitative study. Any survey to be undertaken will require development and access to anonymised databases of cross-border birth seekers.

## Conclusion

Put together, motivations for childbirth abroad vary and evolve. Globalisation, a broken health system, and sustained economic challenges in Nigeria will continue to motivate the middle and upper class of the country to seek childbirth abroad, although the desire is not borne out of desperation. However, while the experiences suggest that it is greener on the other side, it is not necessarily dark green. Systems are needed to elevate the voices of women who seek care abroad and safeguard them from bad care experiences abroad.

## Supporting information

**S1 File. Standards for Reporting Qualitative Research (SRQR).**
(DOCX)

**S2 File. Recruitment advert.**
(DOCX)

**S3 File. Topic guide for in-depth interviews.**
(DOCX)

**S4 File. Inclusivity in global research.**
(DOCX)

## Acknowledgments

We are incredibly grateful to all the women who participated in this study, who gave their time and shared their experiences with us. Without them, the study would have been impossible. We also sincerely thank Aline Semaan, who guided the team in using Dedoose to aid qualitative data analysis and Dr Abimbola Olaniran for critical discussions that informed this paper.

## Author Contributions

**Conceptualization:** Aduragbemi Banke-Thomas, Olayinka Lewis.

**Data curation:** Aduragbemi Banke-Thomas, Olayinka Lewis, Adeola Duduyemi, Olakunmi Ogunyemi, Teeroumanee Nadan.

**Formal analysis:** Aduragbemi Banke-Thomas, Olayinka Lewis, Adeola Duduyemi, Olakunmi Ogunyemi, Teeroumanee Nadan.

**Investigation:** Aduragbemi Banke-Thomas, Olayinka Lewis, Adeola Duduyemi.

**Methodology:** Aduragbemi Banke-Thomas, Olayinka Lewis, Adeola Duduyemi.

**Project administration:** Aduragbemi Banke-Thomas, Olayinka Lewis, Olakunmi Ogunyemi, Teeroumanee Nadan.

**Resources:** Aduragbemi Banke-Thomas, Olayinka Lewis.

**Software:** Aduragbemi Banke-Thomas.

**Supervision:** Aduragbemi Banke-Thomas, Olayinka Lewis.

**Validation:** Aduragbemi Banke-Thomas, Olayinka Lewis.

**Visualization:** Aduragbemi Banke-Thomas.

**Writing – original draft:** Aduragbemi Banke-Thomas.

**Writing – review & editing:** Aduragbemi Banke-Thomas, Olayinka Lewis, Adeola Duduyemi, Olakunmi Ogunyemi, Teeroumanee Nadan.

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
