## [Decision Letter · Decision Letter 0]

30 May 2024

PGPH-D-23-02483

Motivations and experiences of Nigerian women who delivered children abroad: A qualitative study

Dear Dr. Banke-Thomas,

Thank you for submitting your manuscript to PLOS Global Public Health. After careful consideration, we feel that it has merit but does not fully meet PLOS Global Public Health’s publication criteria as it currently stands. Therefore, we invite you to submit a revised version of the manuscript that addresses the points raised during the review process.

As the reviewers have indicated, the manuscript is interesting and focuses on less studied subject. However, the manuscript requires a major revision. The authors are invited to address the concerns raised and resubmit the revised manuscript.

We look forward to receiving your revised manuscript.

Kind regards,

Sunday Adedini, PhD

Academic Editor

Journal Requirements:

2. We have noticed that you have uploaded Supporting Information files, but you have not included a list of legends. Please add a full list of legends for your Supporting Information files after the references list.

Additional Editor Comments (if provided):

As the reviewers have indicated, the manuscript is interesting and focuses on less studied subject. However, the manuscript requires a major revision. The authors are invited to address the concerns raised and resubmit the revised manuscript.

Reviewers' comments:

Reviewer's Responses to Questions

**Comments to the Author**

1. Does this manuscript meet PLOS Global Public Health’s publication criteria? Is the manuscript technically sound, and do the data support the conclusions? The manuscript must describe methodologically and ethically rigorous research with conclusions that are appropriately drawn based on the data presented.

Reviewer #1: Yes

Reviewer #2: Yes

2. Has the statistical analysis been performed appropriately and rigorously?

Reviewer #1: N/A

Reviewer #2: N/A

3. Have the authors made all data underlying the findings in their manuscript fully available (please refer to the Data Availability Statement at the start of the manuscript PDF file)?

Reviewer #1: No

Reviewer #2: No

4. Is the manuscript presented in an intelligible fashion and written in standard English?

Reviewer #1: No

Reviewer #2: Yes

5. Review Comments to the Author

Reviewer #1: Thank you for giving me the opportunity to review this paper. I think it is interesting, with an interesting subject, that I haven’t read anything about before. The manuscript is quite easy to read and well-organized and almost written according to SRQR checklist (se my comments). However, it is not ready for publication and needs quite a lot of editing. It also needs language editing.

Overall, you use deliver and give birth interchangeably in the paper. Women give birth (they do not deliver babies) so please stick to “Nigerian women who have given birth” (or equally), and births instead of deliveries (in the whole paper including table).

In the introduction some topics can be made clearer for the reader:

- You write “At one airport in the UK, the government claimed that during a two-year period, immigration officials stopped over 300 expectant mothers with pregnancies too advanced to be put back on planes to fly back home.” To me it’s not clear whether these mothers are actually birth tourists, from your explanation it seems as though they opted to go back home but couldn’t?

- You also write ”More recently, in Canada, a retrospective analysis of 102 women identified through a Central Triage system as women who delivered in Calgary over an 18-month period found that most were from Nigeria (24.5%).” From your explanation I can’t see that this is actually about birth tourism.

- Several sentences are very long (>50 words) and needs to be shortened or split in two.

- Your aim is to “understand the motivations and experiences of pregnant Nigerian women who travelled out of their country to give birth in another country” and you write that there is only one other study focusing on women’s experiences of birth tourism, but no other reflection is made. I really miss info on what those women experienced as your paper is qualitative and about experiences.

Your objective could be made clearer like this: Our objective of this study is to explore the motivations and experiences of pregnant Nigerian women who travel out of Nigeria to give birth in another country.

Study setting needs to be proofread. Also, the sentence “As established already, Nigeria is one of the top countries of origin of women who seek childbirth abroad, and indeed, giving birth has been described as a trend in the country in recent years.” is missing some word?

Recruitment needs to be further explained, what group do you refer to here: “Through the

advert accessed via a web link posted in the group,”

How is IDIs ideal for maintaining confidentiality? If you are only striving for confidentiality, then I would use an anonymous questionnaire with open ended questions. Wasn’t your aim with IDIs to have depth in your data?

How was the interview-guide “pre-tested” (pilot tested)?

How many interviews did each author perform? How could the women request who was going to perform the interview? In the Author contributions you write that all authors were involved in the interviews but in the methods section only two?

Were the interviews conducted using video call or only audio? Were videos or only audio recorded?

In the methods section you write that collection of data continued until thematic saturation was achieved, however in the discussion you write “However, we continued data collection beyond when data saturation was achieved and continued with additional IDIs as long as we had an opportunity to recruit Nigerian pregnant women to the study.” Explain and make this clear in the method section, please.

I really like that you confirmed your findings with the interviewees, I’d call it “member -checking”

I would strongly recommend you use an additional reference to the Braun and Clarke reference from 2006. They have developed the method a lot since then and published a lot on the developments (as an example they call their method “Reflexive thematic analysis” now).

The analysis need to be further explained, it is very brief now and does not explain the process, how were categories/themes developed, who were involved and so on (see the SRQR checklist)

I don’t understand how the sex of the babies are important for the findings? I would delete that result.

The themes seem underdeveloped, Braun and Clarke explains it as “theme captures an aspect of patterned meaning in the data and tells the reader something about the shared meaning within it, whereas a topic summary simply summarizes participant’s responses relating to a particular topic”. To me these themes are topic summaries. I think you can dig deeper into the data and find better themes.

The quotes often outnumber the describing text in the results which makes me as a reader think the interviews are under-analyzed. As the quotes are often very long and also stacked on top of each other, I recommend you describe more in the text and delete at least half of the quotes (and shorten them).

I also think the discussion is interesting but very long and a bit wordy. Try to prioritize on what to include there.

Reviewer #2: Comments for PGPH-D-23-02483

Motivations and experiences of Nigerian women who delivered children abroad: A qualitative study

The manuscript examines an important but little-studied topic, the issue of travelling abroad to deliver children among Nigerian women. It provided interesting and relevant information concerning the issue of birth tourism in Nigeria. Having gone through the manuscript, I have the following suggestions to improve its quality.

Abstract

I think the conclusion section should contain one or two recommendations for action.

Introduction

This is adequate as the necessary parts of the introduction can be seen (background, problem and objective).

Methodology

In the data analysis section, the authors should specify the exact type of thematic analysis used.

Discussion

The discussion section is adequate. The authors should add a separate conclusion section after the discussion. The paragraph on study strengths and limitations should be given a heading to make it stand out from the rest of the discussion section.

6. PLOS authors have the option to publish the peer review history of their article (what does this mean?). If published, this will include your full peer review and any attached files.

**Do you want your identity to be public for this peer review?** For information about this choice, including consent withdrawal, please see our Privacy Policy.

Reviewer #1: No

Reviewer #2: No

---

## [Decision Letter · Decision Letter 1]

29 Aug 2024

Motivations for and experiences of childbirth abroad amongst Nigerian women: A qualitative study

PGPH-D-23-02483R1

Dear Dr. Banke-Thomas,

We are pleased to inform you that your manuscript 'Motivations for and experiences of childbirth abroad amongst Nigerian women: A qualitative study' has been provisionally accepted for publication in PLOS Global Public Health.

Best regards,

Julia Robinson

Executive Editor

Reviewer Comments (if any, and for reference):

Reviewer's Responses to Questions

**Comments to the Author**

1. If the authors have adequately addressed your comments raised in a previous round of review and you feel that this manuscript is now acceptable for publication, you may indicate that here to bypass the “Comments to the Author” section, enter your conflict of interest statement in the “Confidential to Editor” section, and submit your "Accept" recommendation.

Reviewer #2: All comments have been addressed

2. Does this manuscript meet PLOS Global Public Health’s publication criteria? Is the manuscript technically sound, and do the data support the conclusions? The manuscript must describe methodologically and ethically rigorous research with conclusions that are appropriately drawn based on the data presented.

Reviewer #2: Yes

3. Has the statistical analysis been performed appropriately and rigorously?

Reviewer #2: N/A

4. Have the authors made all data underlying the findings in their manuscript fully available (please refer to the Data Availability Statement at the start of the manuscript PDF file)?

Reviewer #2: Yes

5. Is the manuscript presented in an intelligible fashion and written in standard English?

Reviewer #2: Yes

6. Review Comments to the Author

Reviewer #2: I would like to commend the authors for the review they did of the manuscript and addressing the necessary comments raised in the previous draft.

From my point of view, the manuscript is fine as is and acceptable for publication.

7. PLOS authors have the option to publish the peer review history of their article (what does this mean?). If published, this will include your full peer review and any attached files.

**Do you want your identity to be public for this peer review?** For information about this choice, including consent withdrawal, please see our Privacy Policy.

Reviewer #2: No
